# How Does Cultivar, Maturation, and Pre-Treatment Affect Nutritional, Physicochemical, and Pasting Properties of Plantain Flours?

**DOI:** 10.3390/foods10081749

**Published:** 2021-07-29

**Authors:** Patchimaporn Udomkun, Cargele Masso, Rony Swennen, Bhundit Innawong, Amos Alakonya, Apollin Fotso Kuate, Bernard Vanlauwe

**Affiliations:** 1International Institute of Tropical Agriculture (IITA), Bujumbura 1893, Burundi; 2IITA, Messa, Yaoundé 2008, Cameroon; C.Masso@cgiar.org (C.M.); A.Fotso@cgiar.org (A.F.K.); 3IITA, Kampala 7878, Uganda; R.Swennen@cgiar.org; 4Department of Biosystems, KU Leuven, 3001 Heverlee, Belgium; 5Department of Food Technology, Faculty of Engineering and Industrial Technology, Silpakorn University, Nakhon Pathom 73000, Thailand; b.innawong@gmail.com; 6International Maize and Wheat Improvement Center (CIMMYT), Texcoco 56237, Mexico; A.Alakonya@cgiar.org; 7IITA, Nairobi 30772-00100, Kenya; B.Vanlauwe@cgiar.org

**Keywords:** flour characteristics, plantain-based product, plantain hybrids, pre-treatment, ripening stage, starchy banana

## Abstract

The effect of cultivar, ripening stage, and pre-treatment method were investigated on the nutritional, physicochemical, and pasting properties of plantain flours from two plantains and two plantain hybrids. There were significant variations (*p* < 0.05) in chemical composition and physical properties influenced by the interaction of cultivars, ripening stages, and pre-treatment methods. The highest levels of amylose, water-holding capacity (WHC), and oil-holding capacity (OHC) were observed in unripe flours and acid-treated flour recorded the highest content of resistant starch (RS). Flour after pre-blanching contained the highest level of total phenolic (TP), carotenoid contents, and browning index (BI) value. In contrast, acid-treated flours had the lowest BI value. As ripening progressed, peak viscosity and breakdown values increased but final viscosity, setback, and pasting temperature values were reduced. Untreated flour samples showed the highest peak viscosity. Higher breakdown values were found in acid-treated samples and higher setback values in pre-blanched samples.

## 1. Introduction

Plantains are a multipurpose crop with great processing potential and a source of starch and energy in sub-Saharan Africa (SSA) [1]. They are rich in nutrients such as dietary fiber, minerals (potassium, phosphorus, calcium, sodium, and magnesium), vitamins (A and C), and phenolic compounds (carotenoids and flavonoids) [2,3,4,5]. The availability and concentration of these bioactive compounds vary according to cultivar, postharvest maturation, growing location, climate, agricultural practices, and processing methods [6,7,8]. With approximately 60% of worldwide production [9], the main plantain producing regions are in Central (42%) and West Africa (40%). After increasing urbanization, total production increased by 236% in Central Africa between 2003 and 2017 and 42.2% in West Africa (Figure 1). Countries in Central Africa producing the most plantain are the Democratic Republic of Congo and Cameroon (Figure 1A); Ghana and Nigeria are primary producers in West Africa (Figure 1B).

Although demand in SSA has increased in recent years, plantain production is still seasonal, spanning from a period of plenty to short supplies [10]. Postharvest losses due to poor transportation and distribution facilities in the production areas, harvesting at maturity close to fruit ripening, and poor storage conditions have been reported as constraining factors in production and market chains [11,12]. In general, plantain at the ripe stage can be eaten fresh as an energy-rich food [13]. Green or unripe fruits are mostly preserved as chips or used in composite flours [10,14]. Many studies have reported that flour and starch from unripe plantain are a potential source for healthy functional products as they mainly contain 47–57% dry weight (d.w.) of resistant starch (RS), particularly type II (RS2) and 7–17% fibrous non-starch polysaccharides [15,16]. In the gastrointestinal tract, RS undergoes fermentation in the colon [17] and could resist digestion by α-amylase and glucoamylase. This metabolism increases short-chain fatty acids in the colon [15] and possesses prebiotic potential as it stimulates healthy gut microflora [18]. Therefore, the consumption of flour/starch from unripe plantain as it is or incorporated into food products such as pasta, noodles, infant foods, and bakery products could prevent certain metabolic diseases [19,20]. Moreover, flour/starch could be used as an ingredient in the formulation of gluten-free products for celiac patients [21]. 

The flour of unripe plantain is generally prepared by cutting the peeled fruit into small pieces and air drying for 1–3 days, depending on size and drying methods [22]. Flour obtained by the traditional method is light to dark brown due to enzymatic browning activity. Hence, some pretreatment methods have been proposed to lighten the color, such as blanching and/or soaking in sodium metabisulfite or organic acids [12,23]. Among the organic acids generally regarded as safe (GRAS), ascorbic acid, citric acid, and lactic acid have been chiefly applied [14,24]. After the pre-treatment process, sun-drying alone or combined with hot air drying is used to preserve the flour due to its simplicity and affordability.

Although there is some information available on the physicochemical properties of starches from the widely grown commercial banana [14,25,26], very little is known about starches from plantain and plantain hybrids. To increase productivity and utilization of plantains, proper specifications for harvesting and pulp processing are needed to obtain high-quality flour. Therefore, this study was undertaken to define more fully the effect of ripening stages, pre-treatment processes, and their interactions on the nutritional, physicochemical, and pasting properties of starches from various plantain cultivars and hybrids.

## 2. Materials and Methods

### 2.1. Sample Preparations

Two plantain cultivars (Red Essong and Mbouroukou 3) were collected from Ntui (latitude 4°26.330, longitude 11°34.904, altitude 571 masl, an average temperature of 24.8 °C, and an average rainfall of 1573 m^3^), Cameroon. Two plantain hybrids (PITA 14 and PITA 27) were harvested at full ripening stage with deep green undamaged fruits from a field plot at the IITA station, Yaoundé (latitude 3°51.946, longitude 1°27.735, altitude 750 masl, an average temperature of 23.8 °C, and average rainfall of 1500 m^3^), Cameroon. Red Essong has a French bunch type, whereas Mbouroukou 3 has a False Horn bunch. Producers use both varieties. The presence of both male and female flowers at maturity is found in French plantain. The False Horn bunch has only female flowers and much bigger fruits than the French bunch [27,28]. PITAs are plantain hybrids of the French bunch type, which have resistance to black leaf streak disease and other pests [29]. Though the hybrids are mostly in use under research settings in Yaoundé, some of them have been introduced to farmers in the Center region of Cameroon, including Ntui. The two field sites were established on ferraltic and hydromorphic soils (sandy-clay, clay). The average main soil physic-chemical properties were pH 5.8, organic matter 1.6%. It was abundant in potassium (K) 0.3 ± 0.0 Cmol kg^−1^ soil, it had total available phosphorus (P) of 7.4 ± 2.3 μg g^−1^ soil, total available percent nitrate-nitrogen (NO_3_-N) of 0.1%, and the CaCO_3_ content was 6.1 ± 0.5 Cmol kg^−1^. There was no significant difference between the two sites. Therefore, a randomized complete block design (RCBD) with four replications per cultivar was performed. The size of each plot was 10 × 10 m^2^. The planting distance was 2 m between plants and 2 m between rows, giving a planting density of 2500 plants ha^−1^. The same fertilizers were applied following organic agricultural technology recommendations.

In this study, three stages of maturation were used to produce flour, according to the peel’s external color: 1 = entirely green; 2 = green with a trace of yellow; and 3 = more green than yellow. To attain each stage, the fruits were allowed to ripen naturally in a well-aerated room. Fingers from different bunches were randomly picked at each ripening stage, and fruits were manually cleaned, hand-peeled with a stainless knife, and sliced with a dicer into circular discs with a thickness of approximately 3 mm. Slices were treated in three groups: (i) immersed in water containing 10 g L^−1^ citric acid for 10 min [26]; (ii) blanched at a temperature of 100 °C for 1 min [30]; and (iii) untreated (control). The mixture containing acid-pretreated and sliced pulp was rinsed with water, drained, and blotted with absorbent paper; the blanched sample was only drained and blotted to remove surface water.

After pre-treatment, the plantain slices (500–550 g) were evenly spread on a rectangular stainless steel tray and dehydrated by air convection at a temperature of 70 ± 2 °C using a laboratory-scale hot air dryer (LABEC, Laboratory Equipment Pty Ltd., New South Wales, Australia). The drying experiments were carried out until constant weight was achieved corresponding to a moisture content of 11.0 ± 0.05 g water 100 g^−1^ wet basis (w.b.) with a water activity (a_w_) at 0.5 ± 0.05. The dried slices were ground with a lab grinder (model VMO109, Vita-Mix Corp., Cleveland, OH, USA) for 2 min and then sieved using 60 mesh sieves (ASTM: 60, 250 μm) prior to collection and storage in a 250 g aluminum foil bag at room temperature for further analysis.

### 2.2. Proximate Analysis

The moisture content of the flour samples was determined using the method of AOAC [31]. The sample was dried at 105 °C for 16 h in a hot air oven (model UF55, Memmert Oven, Buechenbach, Germany). Results were expressed in g water 100 g^−1^. Water activity was measured by equilibrating the samples for 20 min using a ventilated hygrometer (model AW-DIO, Rotronic, Bassersdorf, Switzerland) in a thermostatic cell at 25 °C. Protein content was analyzed by the Kjeldahl method. A factor of 6.25 was used to convert from total nitrogen to percentage crude protein [32]. Ash content was investigated by the method of AOAC [31] that involved burning off moisture and all organic constituents at 600 °C in a VULCAN™ furnace (model 3-1750, Cole-Parmer, Vernon Hills, IL, USA). The weight of the residue after incineration was recorded as the ash content. The fat content of the samples was also determined by the method of AOAC [31], using the Soxhlet extraction method (model FOSS Soxtec^TM^ extraction, Hoganas, Höganäs, Sweden). Crude fiber content was recorded using fiber extraction equipment (model FOSS Fibertec^TM^ 2010, Hoganas, Sweden). Carbohydrate content was calculated by subtracting the percentages of moisture, crude protein, ash, fat, and crude fiber from 100. All measurements per block were performed in triplicate (*n* = 12).

### 2.3. Starch Analysis

Amylose content was determined by the iodine binding as described by Williams et al. [33]. A sample of 0.1 g was weighed into a 100 mL conical flask and dissolved with 1 mL of 95% ethanol. Subsequently, 9 mL of 1 N NaOH was added to hydrolyze the starch. The flask was transferred to a water bath and boiled for 10 min, then removed; distilled water was added to make up to 100 mL. Five milliliters were taken from the 100 mL into another conical flask and 1 mL of acetic acid was pipetted into each and 2 mL iodine solution was added to change the color. Distilled water was added to make up to 100 mL and absorbance was read at 620 nm in the spectrophotometer (model UVmini 1240, Shimadzu Corporation, Tokyo, Japan). Potato amylose (Type III, Sigma Chemical Co., St. Louis, MO, USA) was used as standard amylose. Amylopectin content was calculated by subtraction of the percentage of amylose from 100.

The RS content was determined according to the methodology of AACC [34]. Briefly, the protein in the samples was removed by pepsin (40 °C, 1 h, pH 1.5). Then the RS was solubilized and hydrolyzed to glucose by the combined action of α-amylase (Novo Nordisk, Copenhagen, Denmark) and amyloglucosidase (Sigma, St. Louis, MO, USA) for 16 h at 37 °C. The reaction was terminated by adding ethanol (50% v v^−1^) and the RS was recovered as pellets by centrifugation. The RS in the pellet was dissolved in 2 M KOH by vigorous stirring in an ice-water bath for 20 min. This solution was neutralized with acetate buffer and the starch was quantitatively hydrolyzed to glucose with amyloglucosidase. Then free glucose was determined using glucose oxidase enzymes (GOD, Sigma, St. Louis, MO, USA) and peroxidase (POD, Sigma, St. Louis, MO, USA). The RS was calculated as mg of free glucose × 0.9, where 0.9 is the correction factor of glucose-polysaccharide.

Total starch content (TS) was also assessed following the methodology of AACC [34]. The samples were hydrolyzed with a thermostable α-amylase (Novo Nordisk, Copenhagen, Denmark) at 100 °C for 20 min into soluble branched and unbranched maltodextrins. Then this mixture was treated with amyloglucosidase (Sigma, St. Louis, MO, USA) at 60 °C for 45 min. The total released glucose was measured after reaction with glucose oxidase (GOD) enzymes (Sigma, St. Louis, MO, USA) and peroxidase (POD) enzymes (Sigma, St. Louis, MO, USA). Total starch was calculated as mg of free glucose × 0.9 and available starch (AS) as the difference between TS and RS. All analyses per block were performed in triplicate (*n* = 12).

### 2.4. Water-Holding Capacity (WHC)

The method of Anyasi et al. [14] was slightly modified to analyze the WHC of plantain flour. About 1 g flour was mixed with 10 mL of distilled water for 5 min. Subsequently, samples were thoroughly wetted and allowed to incubate at room temperature for 1 h prior to centrifugation at 3000× *g* for 20 min (model Z326K, Hermle Labortechnik, Wehingen, Germany). The supernatant was decanted and the centrifuge tube containing sediment was weighed. The WHC was calculated as g wet residue per g dry flour. The analysis per block was done in triplicate (*n* = 12). 

### 2.5. Oil-Holding Capacity (OHC)

The method described by Savlak et al. [35] was modified. Approximately 1 g of flour was weighed into a 50 mL test tube. Samples were mixed with 25 mL olive oil for 2 min prior to incubation at room temperature for 1 h. Subsequently, the mixture was centrifuged at 3000× *g* for 20 min (model Z326K, Hermle Labortechnik, Wehingen, Germany). After centrifugation, the supernatant was decanted and the centrifuged sample was weighed. The OHC was calculated as g oil per g dry flour sample. The analysis per block was done in triplicate (*n* = 12).

### 2.6. Total Phenolic Content (TPC)

For the first extraction, approximately 1 g flour was mixed with 10 mL of 70% methanol for 10 sec before sonication for 10 min. The sample was then filtered through Whatman No. 4 paper to obtain a clear solution. In the second extraction, 10 mL of 70% methanol was added over the residue, then the sample was homogenized on a vortex and dispensed in a water bath at 80 °C for 5 min. The homogenized sample was filtered into a volumetric flask and extracting solution was added up to a final volume of 25 mL. The extracted sample was then stored at −20 °C prior to analysis.

The TPC was evaluated using the Folin–Ciocalteu assay as described by Shamla and Nisha [36] with a slight modification. A total of 400 μL of extracted sample was mixed with 8 mL of distilled water and 0.5 mL of 2 N Folin–Ciocalteu reagent in a test tube. The mixture was shaken continuously and allowed to react for 6 min. Subsequently, the sample was mixed with 1.5 mL of 20% (w v^−1^) sodium carbonate solution and incubated in a water bath at 40 °C for 30 min. The TPC was measured at 765 nm using a UV-VIS spectrophotometer (model UVmini 1240, Shimadzu Corporation, Japan). A standard calibration curve was plotted using gallic acid. The TPC was expressed as mg of gallic acid equivalent (GAE) per g of sample and analysis per block was done in triplicate (*n* = 12).

### 2.7. Total Carotenoid Content

The method of Rodriguez-Amaya and Kimura [37] was applied. Approximately 3 g of each flour sample plus 3 g of celite were weighed. Successive additions of 25 mL acetone were done to obtain a paste, which was transferred to a sintered funnel (5 µm) coupled to a 250 mL Buchner flask and filtered under vacuum. This procedure was repeated three times until the sample became colorless, and the extract was transferred to a 500 mL separation funnel containing 40 mL of petroleum ether. The acetone was removed through the slow addition of ultrapure water (Millipore) to prevent emulsion formation. The aqueous phase was discarded, and this procedure was repeated four times until no residual solvent remained. The extract was then transferred through a funnel containing 15 g of anhydrous sodium sulfate and made up to a volume of 50 mL with petroleum ether. Samples were read at a wavelength of 450 nm using a UV-VIS spectrophotometer (model UVmini 1240, Shimadzu Corporation, Japan). Total carotenoid content was calculated in μg per g sample. Analysis per block was done in triplicate (*n* = 12). 

### 2.8. Colorimetric Measurement

Color measurements were made using a Hunterlab Spectrophotometer (Hunter Lab Mini Scan Plus Colorimetric, Reston, VA, USA). The color parameters were defined using the CIE L*a*b* color system, where L* is a measure of lightness (L* = 0 for complete black, L* = 100 for perfect white), a* is an indicator of greenness-redness (−a* = greenness, +a = redness), and b* is an indicator of blueness-yellowness (−b* = blueness, +b = yellowness). Before the measurement, the colorimeter was calibrated with a standard white plate (L* = 53.44, a* = −29.94, b* = 12.94). Chroma (C*) or color saturation and browning index (BI) were calculated as follow:(1)$$C*=\sqrt{{a*}^{2}{+\;b*}^{2}}$$
(2)$$\mathrm{BI}=\frac{100\;(x-0.31)}{0.17}$$
where x is obtained using the following equation proposed by Khoozani et al. [38]:(3)$$x=\frac{a^{*}+\;1.75L^{*}}{5.645L^{*}+a^{*}-3.012b^{*}}$$

Three readings were made per sample.

### 2.9. Pasting Properties

Pasting characteristics were determined using a Rapid Visco Analyzer (Model RVA 4500, Newport Scientific, Warriewood, Australia). Three grams of flour were weighed into a previously dried canister and 25 mL of distilled water was dispensed into the canister containing the sample, based on the moisture content. The suspension was thoroughly mixed, and the canister was fitted into the Rapid Visco Analyser. Each suspension was kept at 50 °C for 1 min and then heated up to 95 °C with a holding time of 2 min, followed by cooling to 50 °C with 2 min of holding time. The rates of heating and cooling were at a constant rate of 11.85 °C min^−1^. Peak viscosity, trough, breakdown, final viscosity, and set back as well as peak time and pasting temperature were read from the pasting profile (Figure 2) with the aid of thermocline for Windows software connected to a computer. Analysis per block was performed in triplicate (*n* = 12).

### 2.10. Data Analysis

The effects of different cultivars, ripening stages, and pre-treatment methods on the nutritional, physicochemical, and pasting properties of flours were subjected to statistical analyses using the General Linear Model Program (GLM). Least Squares Means was used to estimate the differences among the means of each factor at 5% of the probability level using the SAS program (version 9.4, SAS, 2002). Mean, standard deviation (SD), coefficient of variation (CV), and standard error (SE) values were also calculated. In addition, the effects of the different treatments and their interactions were compared using the standard error of the difference (SED) of the mean. 

## 3. Results

### 3.1. Proximate Composition

The result revealed that cultivars (C) and their interactive effects with ripening stages (R) and pre-treatment methods (P) had a significant influence (*p* < 0.05) on all proximate compositions of flour samples except moisture content (Table 1). At the same ripening stage and pre-treatment method, the levels of crude protein and total ash contents in the French plantain Red Essong were higher by approximately 14.0 and 12.4% than the corresponding values in the False Horn Mbouroukou 3 while the level of crude fiber was lower by 27.5% (Appendix A). With the same French type of bunch, Mbouroukou 3, had higher crude protein content and total ash than the plantain hybrids PITA 27 and PITA 14, whereas PITA 27 was better in crude fiber content. 

The ripening stages (R) had a significant effect (*p* < 0.05) on all proximate compositions except moisture, carbohydrate, and total ash content, while pre-treatment methods (P) insignificantly affected (*p* > 0.05) crude fat content (Table 1). The control sample (untreated) contained the highest content of crude fat, crude protein, and total ash, followed by acid-treated and pre-blanching flours (Appendix A). An increase in crude fiber was found in acid-treated flour; the pre-blanching process displayed the opposite. Furthermore, the interactive effect of the ripening stage and pre-treatment methods (R × P) was significant (*p* < 0.05) on moisture, crude protein, carbohydrate, and total ash contents. The three-way interaction between cultivar, ripening stages, and pre-treatment methods (C × R × P) was significant (*p* < 0.05) for crude protein, carbohydrate, and total ash content.

### 3.2. Starch Composition and Hydration Properties

The results showed that cultivar significantly influenced (*p* < 0.05) starch composition and hydration properties of flours (Table 2). Among the flours with the same pre-treatment and at the same ripening stage the levels of amylose, TS, RS, WHC, and OHC in the French plantain Red Essong were respectively slightly lower than in the False Horn Mbouroukou 3 by approximately 4.0, 3.1, 4.9, 3.0, and 3.4%, whereas the levels of amylopectin and AS were higher (Appendix A). When the same type of bunch is compared, the French plantain Red Essong contained the highest level of amylose, TS, RS, WHC, and OHC followed by hybrids PITA 27 and PITA 14. Interestingly, the level of RS in Red Essong was about 8.1% higher than in PITA 14 and 2.3% higher than in PITA 27 while the levels of WHC and OHC were higher than in both hybrids by about 8.7 and 4.0%. 

In addition, the ripening stage significantly affected (*p* < 0.05) all starch compositions and hydration properties of flours among the same cultivars and pre-treatment methods (Table 2). The interaction between cultivar and ripening stage (C × R) followed the same pattern as in the variation between ripening stages. With an increase in the ripening stage, the level of amylose, TS, RS, AS, WHC, and OHC decreased while the level of amylopectin increased (Appendix A). The RS content of flour at ripening stage 1 was as high as from 49.7 to 56.4% and decreased by approximately 33.9% at stage 3 (Figure 3A). Similarly, the level of WHC, which ranged from 2.0 to 3.3 and OHC ranging between 1.6 and 2.8 g g^−1^ displayed a reduction by 24.8 and 25.8% during the ripening process (Figure 3B).

When the effect of pre-treatment methods is considered, a significant difference (*p* < 0.05) was observed in all starch compositions as well as in the hydrating properties of flours among the same cultivar and ripening stages (Table 2). The interactive effect of C × P was significant (*p* < 0.05) only on amylose, amylopectin, and RS contents whereas the interaction of R × P had a significant effect (*p* < 0.05) on all characteristics. In addition, an insignificant effect (*p* > 0.05) was found for the interaction of C × R × P on TS. Specifically, a higher level of amylose, AS, WHC, and OHC was found in control (untreated), followed by acid-treated and pre-blanching flours (Appendix A). With the same cultivar and ripening stage, the level of RS content in acid-treated flour was higher than in pre-blanching and control samples by about 8.8 and 4.5% (Figure 3A). Considering the effect of all factors on starch compositions and their hydrating properties, the highest levels of amylose, WHC, and OHC contents were observed in untreated unripe Mbouroukou 3 (stage 1) (Figure 3B), while acid pre-treatment of unripe Mbouroukou 3 flour contained the highest level of RS.

### 3.3. Total Phenolic and Carotenoid Contents

A significant difference (*p* < 0.05) was observed on the effect of cultivar on TP and the carotenoid contents of flours across all ripening stages and pre-treatment methods (Table 3). When the type of bunch is compared, the results showed that the TP and carotenoid content in the False Horn Mbouroukou 3 was higher than in the French Red Essong by an average of 20.6 and 35.8% (Appendix A). Interestingly, the TP content was higher by 36.6% in PITA 27 and 25.9% in PITA 14 compared with Red Essong (Figure 4A). However, the opposite was shown in the carotenoid content as Red Essong was approximately higher by 31.0% compared with PITA 14 and 41.3% with PITA 27 (Figure 4B).

The effect of the ripening stage on TP and carotenoid contents can be seen in all cultivars and pre-treatments as its effect was significantly different (*p* < 0.05) (Table 3). When ripening progressed, the level of TP decreased by about 43.4% and carotenoid contents by about 44.6% (Appendix A). In addition, a markedly significant difference (*p* < 0.05) was observed in the effect of pre-treatment methods on TP and carotenoid contents across all cultivars and ripening stages. The highest levels were recorded in all flours with pre-blanching treatment, followed by the acid-treated sample and control (Figure 4A,B). With the pre-blanching process, 85.4% of TP and 89.2% of carotenoid contents were higher than those of control. Considering the interaction across factors, the interactions between C × P, R × P, and C × R × P on TP and carotenoid contents were significant (*p* < 0.05) but the interaction of C × R affected only the carotenoid contents. In this study, the pre-blanching treatment of PITA 14 at ripening stage 1 exhibited the highest content of TP. The highest level of carotenoids was found in Mbouroukou 3 with the same pre-treatment.

### 3.4. Color

There was a significant difference (*p* < 0.05) in lightness (L*), redness (a*), yellowness (b*), chroma (C*), and browning index (BI) of flours from different cultivars across the same ripening stage and pre-treatment method (Table 3). A higher value of L*, b*, and C* was observed in the French Red Essong when compared with the False Horn Mbouroukou 3 at the same ripening stage and pre-treatment method, while a* and BI values were lower (Appendix A). It could be emphasized that the BI value of Mbouroukou 3 was clearly higher than in Red Essong by an average of 8.2%. In addition, most of the L*, b*, and C* values of Red Essong were higher than in both hybrids. With these results, the BI value of Red Essong tended to be lower than in PITA 14 by about 6.9% and by about 5.9% in PITA 27 (Figure 5).

A significant difference (*p* < 0.05) was found in all color characteristics of flours across cultivar, varying ripening stages, and pre-treatment methods (Table 3). With increasing ripening stage, the L* value decreased by 6.3%, while a considerable increase of 10.3% for b*, 10.3% for C*, and 17.3% for BI was presented (Appendix A). The results also showed that all acid-treated samples were notably lighter with a higher L* value than flours from control and pre-blanching, whereas the a*, b*, C*, and BI values were lower. To be more specific, acid pre-treatment resulted in a better color with 17.4% of BI value higher than the control and 25.1% higher than the pre-blanching sample (Figure 5). Moreover, the interactive effects of C × R, C × P, R × P, and C × R × P on all color characteristics were also significant (*p* < 0.05). Lower BI values were obtained for the flour produced from acid pre-treatment of Red Essong at ripening stage 1. The pre-blanching process of Mbouroukou 3 exhibited the darkest color at ripening stage 3.

### 3.5. Pasting Properties 

The pasting properties of flours significantly varied (*p* < 0.05) with cultivar (Table 4). The peak viscosity, trough viscosity, final viscosity, breakdown, and setback values of the French Red Essong were approximately higher than those of the False Horn Mbouroukou 3 by 11.8, 9.2, 9.1, 9.8, and 9.0%, respectively (Appendix A). A greater variation was observed among the French bunch types. Hybrid PITA 27 exhibited the highest value of peak viscosity, trough viscosity, final viscosity, breakdown, and pasting temperature, followed by Red Essong and PITA 14. 

Likewise, all pasting properties indicated statistically significant differences (*p* < 0.05) between ripening stages (Table 4). With increasing ripening, peak viscosity, trough viscosity, and breakdown values, respectively, increased by an average of 13.5, 11.4, and 18.2%, while final viscosity, setback, and pasting temperature values showed the opposite (Appendix A). The interactive effect of C × R on the pasting properties was significant (*p* < 0.05), except on final viscosity and peak time. When the effect of pre-treatment methods is considered, a significant difference (*p* < 0.05) was observed in the pasting properties of the same cultivars and at the same ripening stages. The control sample had the highest peak viscosity, trough viscosity, and final viscosity, followed by acid-treated and pre-blanching flours. A higher breakdown value was found in acid-treated flours when compared to control and pre-blanching samples. Although the pre-blanching process resulted in the highest setback value, its pasting temperature was the lowest. The interactive effect of C × P was significantly different (*p* < 0.05) for all pasting properties, but no significant differences occurred for an interaction between R × P on trough viscosity and final viscosity. The three-way interaction effects between C × R × P were statistically significant (*p* < 0.05) for all properties, except final viscosity and peak time.

## 4. Discussion

The chemical composition changes indicated biochemical reactions and a climacteric peak between the stages [20]. The moisture content of the flours in this study was close to the results reported by Ssonko and Muranga [39] in starches of the East African highland banana (11.1–11.8%). Still, it was higher than the findings of Rayo et al. [40] and Rodriguez-Ambriz et al. [41], who respectively reported 4.0 and 6.0% moisture content in unripe banana flours. For crude protein content, our results agreed with that reported by Kumar et al. [42] for green banana flour from dessert and plantain (3.4–4.9%). Crude fat content was not detected in the East African highland banana [39], but it was the opposite of this study. Our result was similar to that found by Kumar et al. [42], which displayed the crude fat content in a range of 0.17 to 0.61% in green banana flours. Although the ash content was higher than that reported by Eggleston et al. [25] for plantain (0.27–0.34%), plantain hybrids (0.28–0.32%), and cooking banana (0.35–0.41%), it was in a close range to a study of Suntharalingam and Ravindran [43] for green banana flour from Monthan (4.2%) and Alukehel (3.3%). The amylose and amylopectin content of plantain flours were in the same range in many studies. For example, Vatanasuchart et al. [44] exhibited amylose content of 38.6–43.8% for six Kluai cultivars, while amylose content of 23.0–24.2% was obtained for Monthan and Saba cultivars Kumar et al. [42]. The results reported by Utrilla-Coello et al. [45] showed that the amylose content of Musa AAA (19.3–22.0%) was lower than Musa AAB (26.4%) and Musa ABB (25.4%). In addition, the present result most likely agreed with the previous research that green banana contains approximately 70–75% total starch content [17,46].

The ethylene production could explain the reduction of protein content in most cultivars during ripening. Proteins are involved in several metabolic pathways during ripening and the senescence of the fruits [47]. According to Toledo et al. [48], some protein enzymes are synthesized during pre-climacteric and climacteric stages and are involved in developing the flavor and texture of bananas. However, these enzymes are down-regulated, and protein could be used as a source of energy [49]. An increase in fat content of most cultivars might be possible in that conversion of carbohydrate to fat was taking place during the ripening stage, as indicated by Campuzano et al. [49]. Baiyeri et al. [50] assigned an increase of total ash in ripe plantain to the release of mineral elements like potassium and magnesium, caused by tissue breakdown during ripening, but we demonstrated the opposite. Fully green plantain had the highest amylose content compared with that of ripening stage 3. According to Gao et al. [51], a rapid reduction of apparent amylose content in bananas after stage 2 was due to carbohydrate enzymatic hydrolysis. Bi et al. [52] showed that the amylose content remained almost unchanged from ripening stage 1 to 5, although it significantly decreased at ripening stage 7 due to further degradation during full maturation. Liu et al. [53] suggested that a decrease of amylose content was generated by hydrolysis in the amorphous part of the starch granule.

A lower protein and fat content in the treated flour was caused by leaching during the modification process, as indicated by Gutiérrez et al. [54]. It is worth noting that despite the leaching of these nutrients during pretreatment, their values were higher than those reported by Pelissari et al. [55] for plantain starch. This could imply that flour from treated plantain is richer in nutritional components than starch and this has a significant potential for application in gluten-free products for celiac [21]. Likewise, an increase of dietary fiber in acid-treated flours was mainly due to starch modification [56]. This is interesting since a high content of total dietary fiber could have prebiotic effects on the growth of beneficial bacteria in the human colon [57].

Unripe plantain flour is underscored as a potential source of RS that presents high in vitro fermentability [16]. According to our results, all cultivars analyzed, especially unripe Mbouroukou 3 and Red Essong, had a very high RS content of about 44%, which was nearly equal to a value obtained by Pelissari et al. [55] for plantain flour (49.5% RS). Kumar et al. [42] indicated that the ABB group of starchy bananas was more likely to record a higher level of RS than the AAB and AAA groups. It has been previously described that green banana is rich in starch, majorly in the form of RS type II (RS2), which corresponds to native uncooked starch that is barely susceptible to hydrolysis [58]. The RS fraction of banana flour decreased when fully ripe [59] and agrees with this study. Gao et al. [51] also mentioned a gradual reduction of TS and RS during the ripening process in the Cavendish cultivar. Similarly, Wang et al. [60] displayed a rapid decrease in Cavendish’s RS content during the first four stages of ripening. Therefore, the cultivars and ripening stages used could also be suitable for beneficial effects on human health, particularly for lower fasting blood glucose levels and the low-density lipoprotein (LDL) cholesterol/high-density lipoprotein (HDL) cholesterol ratio [38,61,62]. Although plantain flours presented in this study showed a low amount of AS, this could be considered a crucial issue for consuming digestible carbohydrates.

In addition, the susceptibility of starch to enzyme digestion varies depending on the variety as well as processing methods [63]. With the same cultivar and ripening stage, a higher content of RS was found in acid-treated flour when compared with pre-blanching and untreated samples. This is due to hydrolysis of the amorphous zone of the starch with a concomitant increase in the crystallinity and changes in the starch structure, producing chains that are less accessible to digestive enzymes [64]. Almanza-Benitez et al. [61] reported a similar pattern of high RS content in HCl acid-treated unripe plantain flour. A high content of RS in flour with pre-blanching treatment may be partially explained by the less digestible properties of amylose content. However, Zhang and Hamaker [65] suggested that the complexes formed during starch retrogradation after cooking, which were considered to be retrograde RS type III, could potentially contribute to this phenomenon. Giraldo Toro [66] indicated that RS decreased when the extent of thermal gelatinization increased. Therefore, the optimization of process parameters during flour production should be analyzed to produce flour with a high RS content.

Results of WHC and OHC in this study were similar to those reported by Anyasi et al. [14] and Alkarkhi et al. [67] on flours from unripe banana pulp. A higher WHC and OHC in the False Horn Mbouroukou 3 could be attributed to its higher level of amylose and dietary fiber as exhibited in the results. With an increase in ripening stage, a reduction of WHC may be associated with starch degradation and sugar release; possibly sugar could interact with starch chains [68] or water, limiting the availability of water to hydrate the starch [69]. According to Anyasi et al. [14], the efficiency of unripe banana flour in holding water and oil was directly related to amylose and dietary fiber content. Though Campuzano et al. [49] showed that the oil absorption capacity of banana flour increased during ripening, and was correlated to the increase of fat, it was not the case of this study. The hydration properties of plantain and banana flours could also be influenced by various intrinsic factors such as the physical characteristics of starch granules (structure, surface polarity, particle porosity), and the composition and number of amino acids and protein in the flour [37,38,67]. In general, a high WHC of Mbouroukou at ripening stage 1 indicates its potential as a thickener in semi-liquid and liquid foods [70]. Flours with a high OHC property could be applied as an alternative emulsifying agent in food systems with high-fat content, such as bakery products [71].

Plantain contains several bioactive compounds, such as phenolics, carotenoids, biogenic amines, and phytosterols [2,72,73], which are highly desirable in the diet as they exert many positive effects on human health and well-being. Many of these compounds have antioxidant potentials and are effective in protecting the body against various oxidative stresses [74]. There are various classes of phenolic compounds in banana and plantain. Méndez et al. [75] investigated some extractable phenolic compounds (free phenolic acids) in bananas from Tenerife and Ecuador. High free gallic acid and catechin (cianidanol) content were reported in two varieties of bananas. However, banana pulp also contains several different cell non-extractable phenolic compounds [76]. Bennett et al. [7] detected condensed tannins and flavonoids (catechin, gallocatechin, and epicatechin) in the soluble cell wall fractions of the fruit pulp. They showed the presence of anthocyanidin delphinidin in the cell walls. Epicatechin, epigallocatechin, and gallocatechin have been detected in Dwarf Cavendish banana as studied by Harnly et al. [77], while caffeic and chlorogenic acids, kaempferol, and apigenin were the dominant phenolics in Anatolian plantain (a traditional endemic food plant of Eastern Anatolia, Turkey) [78]. In this study, there was a great diversity in the content of TP and carotenoids among the cultivars used. The higher content of TP and carotenoids in the False Horn Mbouroukou 3 than in the French Red Essong and the hybrids might be associated with the effect of cultivar, climate, and soil conditions. This corresponds with Tsamo et al. [79], who demonstrated large diversity of phenolic profiles among cultivars in the pulp of nine plantain cultivars. Udomkun et al. [5] showed a significant effect of cultivar, ripening stage, bunch type, and location on the variability of provitamin A carotenoids (pVACs) in cultivars and hybrids.

Through fruit ripening, a reduction of TP content in flours might be linked to the activity of oxidative enzymes such as polyphenol oxidase (PPO) [80]. In addition, Starrett and Laties [81] indicated that a reduction in the quantity of this compound might be due to a reduction of enzyme activity, such as phenylalanine ammonia-lyase, in its biosynthesis pathway. A reduction of serotonin and dopamine levels which also act as antioxidants in banana cultivars can also be associated with an oxidative pathway activated during fruit ripening [82,83,84], although the declining level of carotenoids as ripening progressed was reported by Ngoh Newilah et al. [85]. They explained the mechanism as the enzymatic cleavage of carotenoids, which may stimulate the synthesis of other carotenoid isomers, such as volatile compounds. All cultivars examined recorded high levels of TP that can be attributed to pretreatment with organic acid that also acts as an antioxidant. Although many studies of retention of TP and carotenoids in different thermal processes indicate that the exposure of samples to heat results in substantial losses of these bioactive compounds [83,85], the findings in this study reported a conflict. High retention of TP and carotenoids in flours after the pre-blanching process could be attributed to damaged cell structures of the food matrix, thus giving access to digestive enzymes and improved bio-accessibility [86].

The color attributes of flour have an impact on both the consumer and the food industries. According to Jeet et al. [87], a brighter color could be more acceptable for consumers. Hence, incorporation as an ingredient of flour from the green unripe French plantain Red Essong is suggested without altering the color of finished products unlike for flours from the False Horn Mbouroukou 3 and both hybrids. The lighter pulp color of Red Essong might be associated with a lower level of carotenoid content. Borges et al. [88] showed that banana and plantain pulps with lighter colors tend to contain a higher level of flavonoid contents. When ripening progressed, the conversion of starch to sugar can be used to explain the darker color of flours [36]. Therefore, the selection of appropriate cultivars and ripening stages is recommended depending on the specific purpose of each processed product. In addition, high L* and low a*, b*, C*, and BI values recorded in all acid-treated samples can be attributed to the partial inhibition of the PPO under acid pH [89]. Similar reports were shown by Anyasi et al. [14] and De Souza et al. [90] on the effect of organic acids on fresh-cut mango and unripe banana flours, respectively.

Furthermore, Gutiérrez et al. [21] explained the loss of the yellow coloration (b* value) in the modified plantain flours because of the leaching of some pigments during processing. In contrast, flours with pre-blanching treatment had a darker color than other samples. Many darkening factors, such as enzymatic oxidation, vitamin C oxidation, Maillard reaction, caramelization, etc., are attributed to the decreased brightness value during thermal processing [91]. Tao et al. [92] also found this in cabbage treated by sequential blanching and drying.

The pasting property provides a basis for applications in industrial processing. Firstly, note that it might be difficult to compare our RVA results with those reported for other flours/starches due to differences in sample concentration and methodology (Brabender viscoamylogram, micro viscoanalyzer) used for the determination of their pasting characteristics. In this study, the pasting properties of flours varied significantly by cultivar, ripening stages, and pretreatment methods. This is similar to Ssonko and Muranga [39], who reported a significant difference in the pasting properties of starches from the different cultivars of the East African highland banana. However, from the application point of view, these differences were marginal and would not affect the application suitability to their pasting properties of interchanging the cultivars. With increasing ripening, peak viscosity increased, and this agrees with Campuzano et al. [49] in Cavendish banana flour. Even though the results in this study differed from the previous study in banana starch [93], they are quite similar to that of Bakare et al. [94], who researched unripe banana flour. These divergences in apparent viscosity might be explained either by varietal reasons or the undefined ripening stage of the flours used. The most plausible explanation for the changes observed might be related to the enzymatic activity during ripening. Gao et al. [51] proposed that β-amylase plays a significant contribution to starch degradation during banana ripening, and α-amylase has a supporting role. de Nascimento et al. [95] also indicated that β-amylase is highly induced during ripening and suggested that its activity was highly correlated to starch degradation. The maximum viscosity during heating (peak viscosity) which was found, particularly in the untreated flour of hybrid PITA 27, suggests that this flour could be used in food formulation where high viscosity is required, whereas the treated flours from both the False Horn Mbouroukou 3 and hybrid PITA 14 could be employed for the foods that require less thickening. Gutiérrez et al. [21] recommended flours with low peak viscosity values for commercial bread and cookie making at an industrial level.

The breakdown is used as an index of the brittleness of starch granules against the shear forces generated during heating [96]. Thus, the higher the breakdown value in the acid-treated flour of PITA 27, the lower the stability of the starch matrix against shear forces during the heating period. Although the setback value is an index for measuring the rate of starch retrogradation and syneresis, the higher value of setback, which was also observed in the flour of stage 1, especially Red Essong and PITA 27 with pre-blanching pretreatment, indicates the maximum tendency of retrogradation. A reduction of setback value during ripening is due to the decrease in the amount of starch able to gelatinize [97]. According to Sackey [98], starch retrogradation causes textural changes, often resulting in unwanted properties such as staling in bread, skin formation, paste gelling, and loss in clarity, which are important attributes for the food processor as well as consumers. A study by Afoakwa and Sefa-Dedeh [91] linked the property of high retrogradation to the high degree of association between the starch-water systems and their high ability to recrystallize. Moreover, the pasting temperature has been described as the temperature above the gelatinization temperature when starch granules begin to swell. It is used as an indicator for the minimum temperature required to cook the flour [99]. The pasting temperature of plantain flour occurred between 60.9 and 90.6 °C. These temperatures were mostly higher than that of cassava starch (69.5 °C), which has a lower amylose content [100]. In this study, the highest pasting temperature was recorded for flour obtained from PITA 27 at ripening stage 1 with/without acid pretreatment. Blazek and Copeland [97] indicated the structural stability of banana starch granules is influenced both by amylose molecules and the degree of crystallinity in amylopectin. As Wahab et al. [100] stated that the maintenance of the gelatinization process at a low temperature will result in improved bread quality, PITA 27 flour which is highly resistant to swelling and rupturing, could be used as a new ingredient in bread baking.

At the processing level, the use of flour obtained from PITA 27 and Red Essong could lead to products that are more stable during storage as it is less susceptible to retrogradation. However, the greater tendency towards retrogradation in the False Horn Mbouroukou 3 and PITA 14 may provide some health benefits. Apart from being a good source of TP and carotenoids, the retrograded starch of these plantains is considered a type of resistant starch. Hence, it may allow the growth of friendly bacteria such as *Bifidobacterium* and *Lactobacillus* [21], which act as prebiotics. Moreover, Hernández et al. [101] reported that the fermentation of retrograded starch in the human colon leads to the production of fatty acids with small molecular weight such as butyric acid that can be absorbed, resulting in the reduction of cholesterol levels in the blood.

As mentioned in the results, a change in all rheological properties was noticed during the ripening process, which is most likely linked to the chemical composition of the flours. Campuzano et al. [49] reported a strong positive correlation between pasting properties and carbohydrate content and a strong, negative correlation between pasting properties and sugar content. Apart from cultivars and ripening stages, the difference in the pasting properties of the studied flours might be associated with the type of processing methods, the microstructure of flour particles, and the variation in the starch quantities, including damage and RS contents [26]. Additionally, Pelissari et al. [55] mentioned that the pasting properties of flours could be influenced by the granular size, the amylose/amylopectin ratio, the volume fraction of suspended solids, the affinity between the hydroxyl groups of molecules, the molecular weight of amylose leached from the starch granules, and the conditions of the thermal process used to induce gelatinization.

## 5. Conclusions

This study provided data on the influence of cultivar, ripening stages, and pre-treatment methods on the chemical and physical properties of two plantain cultivars (French and False Horn bunch types) and two hybrids. Plantains at stages 1‒3 were treated by two different pre-treatment methods (acid and blanching) before making flours. Unripe plantain Mbouroukou 3 and Red Essong had a higher level of RS, WHC, OHC, and carotenoids than the hybrids. However, the two hybrids, especially PITA 27, were better in peak viscosity, trough viscosity, final viscosity, breakdown, and pasting temperature. Considering the nutritional aspects, plantain Mbouroukou 3 and Red Essong are recommended as they could have a beneficial effect on human health comparing to hybrids. However, hybrid PITA 27 represents a source material with great commercial potential, particularly in the bread-making industry, due to its pasting properties. As ripening progressed, higher viscosity and breakdown values were recorded, but the bright color and level of RS, TP, and carotenoid content diminished. When the effects of pre-treatment methods are compared, an improvement of RS and color was clearly observed in acid-treated flours, whereas flours with pre-blanching treatment were significantly higher in bioactive compounds. While our findings may provide comprehensive knowledge for the production of plantain flours, their application as an alternative ingredient in various functional foods should still be explored. In addition, more cultivars should be investigated, promoted, and used in breeding programs for the development of appropriate cultivars that could meet the different requirements of processors and consumers.

## Figures and Tables

**Figure 1 foods-10-01749-f001:**
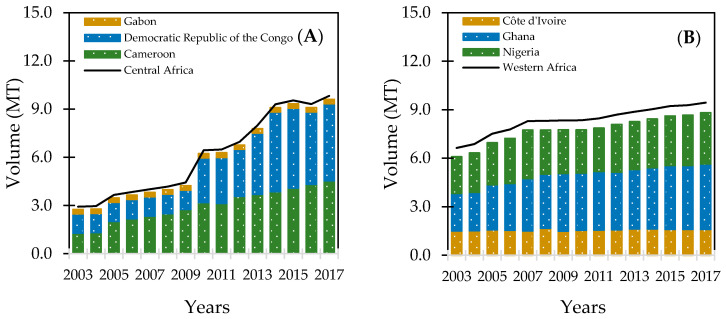
Plantain production in Central (**A**) and West (**B**) Africa. Source: FAOSTAT [9].

**Figure 2 foods-10-01749-f002:**
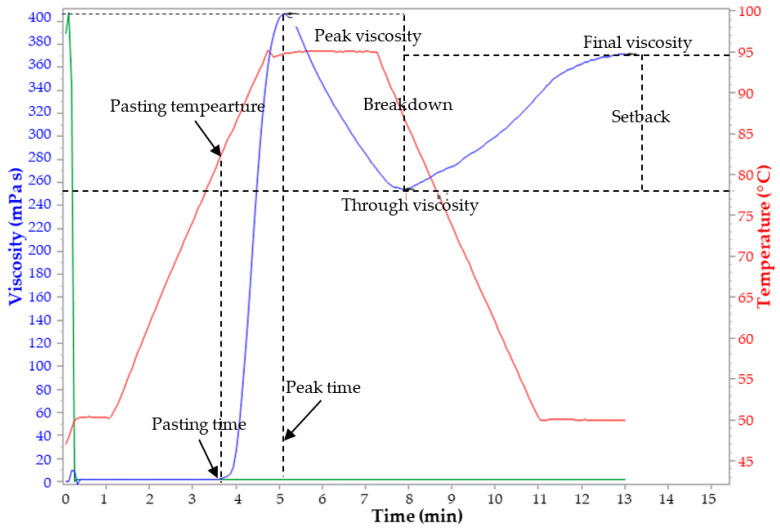
Pasting properties of plantain flour.

**Figure 3 foods-10-01749-f003:**
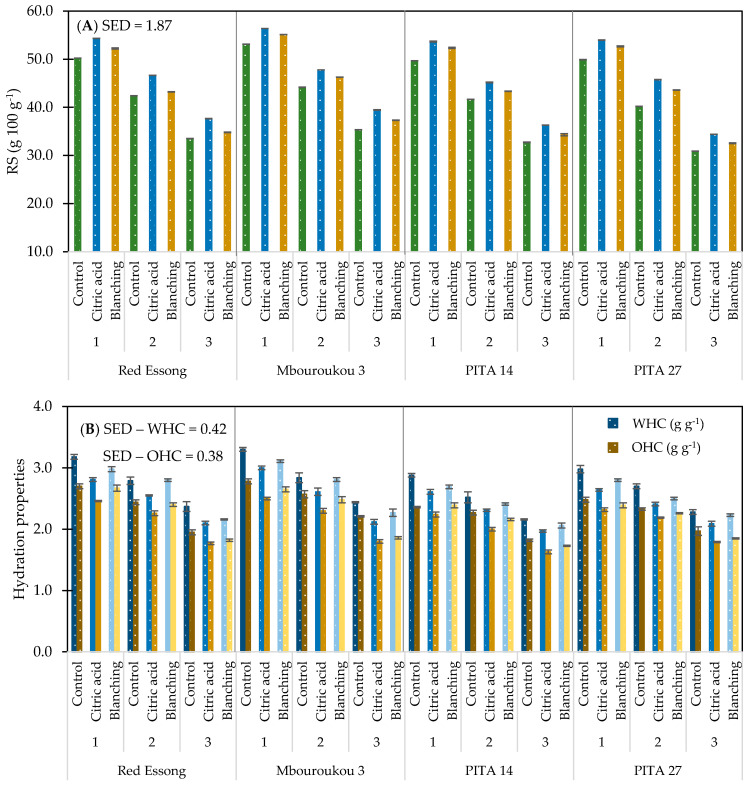
Resistant starch (**A**) and hydration properties (**B**) of plantain flours as influenced by cultivar, ripening stages, and pre-treatment methods. Ripening stage: 1 = mature green; 2 = green with a trace of yellow; and 3 = more green than yellow.

**Figure 4 foods-10-01749-f004:**
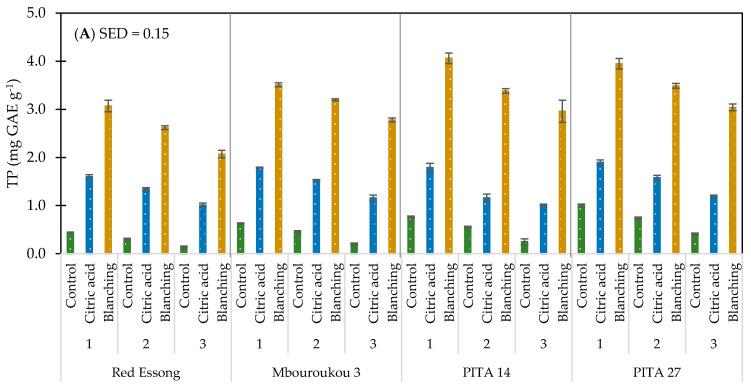
Total phenolic (**A**) and carotenoid contents (**B**) of plantain flours as influenced by cultivar, ripening stages, and pre-treatment methods. Ripening stage: 1 = mature green; 2 = green with a trace of yellow; and 3 = more green than yellow.

**Figure 5 foods-10-01749-f005:**
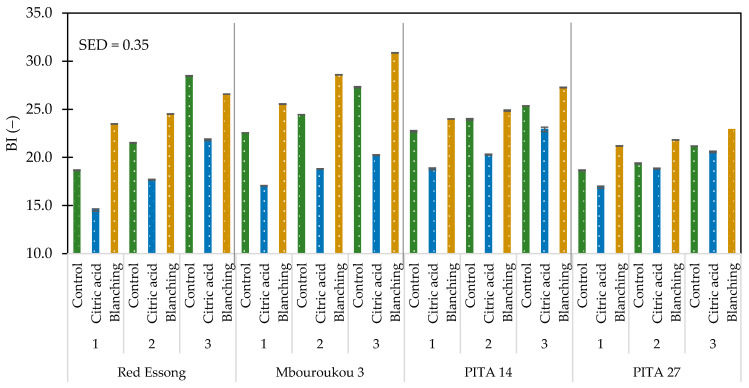
Browning index of plantain flours as influenced by cultivar, ripening stages, and pre-treatment methods. Ripening stage: 1 = mature green; 2 = green with a trace of yellow; and 3 = more green than yellow.

**Table 1 foods-10-01749-t001:** ANOVA statistics showing the effect of cultivar, ripening stages, and pre-treatment methods on the proximate composition of plantain flours.

Parameter	Moisture (g 100 g^−1^)	Crude Fat (g 100 g^−1^)	Crude Protein (g 100 g^−1^)	Carbohydrate (g 100 g^−1^)	Total Ash (g 100 g^−1^)	Crude Fiber (g 100 g^−1^)
Range	10.47–12.20	0.45–0.73	2.92–4.56	79.25–82.09	1.62–3.22	1.34–2.66
Mean	11.22	0.59	3.65	80.57	2.09	1.87
SD	0.39	0.07	0.40	0.79	0.38	0.31
CV	3.47	12.13	10.86	0.99	17.94	16.48
SE	0.05	0.01	0.05	0.09	0.04	0.04
p of cultivar (C)	ns	***	***	***	***	***
p of ripening stage (R)	ns	***	***	ns	ns	***
p of pre-treatment (P)	***	ns	***	***	***	***
p of C × R	ns	***	***	***	***	***
p of C × P	ns	***	***	***	***	***
p of R × P	***	ns	***	***	***	ns
p of C × R × P	ns	ns	***	***	***	ns

*** = significant at *p* < 0.05 and ns = not significant.

**Table 2 foods-10-01749-t002:** ANOVA statistics showing the effect of cultivar, ripening stages, and pre-treatment methods on the starch composition and hydration properties of plantain flours.

Parameter	Amylose (g 100 g^−1^)	Amylopectin (g 100 g^−1^)	TS (g 100 g^−1^)	RS (g 100 g^−1^)	AS (g 100 g^−1^)	WHC (g g^−1^)	OHC (g g^−1^)
Range	22.51–31.25	68.75–77.49	60.23–86.84	30.52–56.48	24.89–34.08	1.95–3.32	1.61–2.80
Mean	26.74	73.31	73.99	43.97	30.02	2.57	2.21
SD	2.24	2.19	8.53	7.66	1.93	0.35	0.31
CV	8.39	2.99	11.53	17.42	6.44	13.58	14.08
SE	0.26	0.26	1.01	0.90	0.23	0.04	0.04
p of cultivar (C)	***	***	***	***	***	***	***
p of ripening stage (R)	***	***	***	***	***	***	***
p of pre-treatment (P)	***	***	***	***	***	***	***
p of C × R	***	***	***	***	***	***	***
p of C × P	***	***	ns	***	***	ns	ns
p of R × P	***	***	***	***	***	***	***
p of C × R × P	***	***	ns	***	***	***	***

*** = significant at *p* < 0.05 and ns = not significant. TS: Total starch content; RS: the highest content of resistant starch; AS: available starch; WHC: water-holding capacity; OHC: oil-holding capacity.

**Table 3 foods-10-01749-t003:** ANOVA statistics showing the effect of cultivar, ripening stages, and pre-treatment methods on the total phenolic, carotenoid contents, and color alterations of plantain flours.

Parameter	TP (mg GAE g^−1^)	Carotenoids (μg g^−1^)	L* (−)	a* (−)	b* (−)	Chroma (−)	Browning Index (−)
Range	0.12–4.13	0.03–3.78	47.09–70.48	0.31–1.19	8.63–14.39	8.66–14.41	14.51–30.94
Mean	1.68	1.13	56.88	0.75	11.07	11.10	22.34
SD	1.19	0.96	4.84	0.18	1.56	1.56	3.72
CV	70.68	84.78	8.50	24.60	14.12	14.07	16.64
SE	0.14	0.11	0.57	0.02	0.18	0.18	0.44
p of cultivar (C)	***	***	***	***	***	***	***
p of ripening stage (R)	***	***	***	***	***	***	***
p of pre-treatment (P)	***	***	***	***	***	***	***
p of C × R	ns	***	***	***	***	***	***
p of C × P	***	***	***	***	***	***	***
p of R × P	***	***	***	***	***	***	***
p of C × R × P	***	***	***	***	***	***	***

*** = significant at *p* < 0.05 and ns = not significant. TP: the highest level of total phenolic; L*: lightness; L* = 0 (black color); L* = 100 (white color); a*: green color (−)/ red color (+); b*: blue color (−)/ yellow color (+).

**Table 4 foods-10-01749-t004:** ANOVA statistics showing the effect of cultivar, ripening stages, and pre-treatment methods on the pasting properties of plantain flours.

Parameter	Peak Viscosity (mPa s)	Trough Viscosity (mPa s)	Final Viscosity (mPa s)	Breakdown (mPa s)	Setback (mPa s)	Peak Time (min)	Pasting Temperature (°C)
Range	219.5–694.7	170.8–487.6	289.3–580.8	14.3–262.6	5.9–226.2	4.8–7.0	60.9–90.6
Mean	445.0	293.8	416.2	153.8	123.1	5.3	80.5
SD	114.6	64.3	68.6	67.8	55.8	0.6	8.2
CV	25.7	21.9	16.5	44.1	45.3	10.3	10.1
SE	13.5	7.6	8.1	11.3	9.3	0.1	1.0
p of cultivar (C)	***	***	***	***	***	***	***
p of ripening stage (R)	***	***	***	***	***	***	***
p of pre-treatment (P)	***	***	***	***	***	***	***
p of C × R	***	***	ns	***	***	ns	***
p of C × P	***	***	***	***	***	***	***
p of R × P	***	ns	ns	***	***	***	***
p of C × R × P	***	***	ns	***	***	ns	***

*** = significant at *p* < 0.05 and ns = not significant.

## Data Availability

This manuscript has no associated data.

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
