# Peer review of "How Does Cultivar, Maturation, and Pre-Treatment Affect Nutritional, Physicochemical, and Pasting Properties of Plantain Flours?"

_foods, 2021, doi:10.3390/foods10081749_

Round 1

Reviewer 1 Report

The manuscript is interesting and a lot of data are presented but it has to be improved in the experimental design, data analyses and presentation of results:

You wrote that plantain cultivars were collected from Ntui and plantain hybrids from Yaoundé: can you specify which experimental design did you use in the field?

You had a single factor experiment, and the treatments consisted of 4 genotypes: 2 cultivars which were cultivated in one site, and 2 hybrids which were cultivated in another site. Therefore I wonder how did you analyze the data. Is it correct to compare data from the 4 treatments in the same Anova, and to compute “p of cultivar” as you did, since the 4 treatment were not in the same site? You could not quantify the genotype*environment interaction, which has to be considered when comparing different genotypes.

May be thet data could be presented genotype by genotype, discussing the effect of the two factors (R and P) in each genotype as differences versus the respective control; or may be that a PCA could allow the reader to more clearly understand the effect of the factors R and P on the 4 genotypes.

A lot of interesting data are presented, but the four Tables are quite hard to read and also the results section is hard to read, following the Tables. I suggest to report the letters to highlight when differences are significant between two means. I also suggest to present the data in a more readable way, may be try to select the more interesting and useful parameters (for example, is it necessary to report L*, a*, b* and then C* and BI, which are a combination of the previous data?). Try to lighten the tables, and to find a clearer way to present and to highlight the most important results.

Discussion is well written and interesting conclusion are reported.

Reviewer 2 Report

The paper looks interesting and probably will find some interest of the readers. The raw material investigated by Authors has a local meaning nut it doesn’t mean is not worth of investigation. There is strong need to understand some phenomena  of ripening of most crops especially in times of climat changing. Unfortunately there are also some points in the paper that needs to be upgraded or explain. The list of them is presented below. According to that I have to designate the paper as needed minor revision.

  1. Please explain the factor for protein content. Any reference for 6.25?
  2. Did you perform any calibration for amylose content?
  3. In my opinion you should also perfom an analysis of molecular parameters of starch including average molecular weight and dispersity factor
  4. As Authors have mentioned there is not to much data on plantain flour but in my opinion the scientific discussion should be done by comparison with any scientific data from literature
  5. The color analysis is OK but I advise to show some pictures as well
  6. The data from Table 4 should be supplemented with appropriate charts
  7. The discussion section is very poor in explanation of phenomena observed. It should be upgraded
  8. You established the total content of phenolics, carotenoids etc. What about the particular compound? I think some analysis should also be done to show which compounds dominates in those fractions

Round 2

Reviewer 1 Report

Line 100 Please add how many replicates in the RCBD, you wrote they are four in the cover letter but you did not in the text.

Anyway, it would have be more sound to growth the two cultivars at the at the IITA station in the 4 blocks with the hybrids. You can not run an anova across 4 genotypes (2 cultivars and two hybrids) since yours is not a complete block design: you do not have the treatment "genotype" in each block, the blocks are incomplete, because you have two cultivars in 4 blocks in Ntui and two hybrids in four blocks in Yaoundè. Even if the two locations are similar, the blocks are incomplete. Therefore you should run two different anova, one to compare the cultivars from Ntui and another anova to compare the two hybrids, if you want to use the four replications as a source of variation in your anova. If not, you probably pooled the fruits from the different blocks and then you planned a completely randomized design in the room to compare all the treatments, with three replicates,  since you wrote that "measurements were performed in triplicate" , then I infere that you did not use the replicates from the blocks which were four. Please explain this aspect. 
